# CRISPR-mediated genetic interaction profiling identifies RNA binding proteins controlling metazoan fitness

Adam D Norris[1,2], Xicotencatl Gracida[1], John A Calarco[1,3]*

[1]FAS Center for Systems Biology, Harvard University, Cambridge, United States; [2]Department of Biological Sciences, Southern Methodist University, Dallas, United States; [3]Department of Cell and Systems Biology, University of Toronto, Toronto, Canada

**Abstract** Genetic interaction screens have aided our understanding of complex genetic traits, diseases, and biological pathways. However, approaches for synthetic genetic analysis with null-alleles in metazoans have not been feasible. Here, we present a CRISPR/Cas9-based Synthetic Genetic Interaction (CRISPR-SGI) approach enabling systematic double-mutant generation. Applying this technique in *Caenorhabditis elegans*, we comprehensively screened interactions within a set of 14 conserved RNA binding protein genes, generating all possible single and double mutants. Many double mutants displayed fitness defects, revealing synthetic interactions. For one interaction between the MBNL1/2 ortholog *mbl-1* and the ELAVL ortholog *exc-7*, double mutants displayed a severely shortened lifespan. Both genes are required for regulating hundreds of transcripts and isoforms, and both may play a critical role in lifespan extension through insulin signaling. Thus, CRISPR-SGI reveals a rich genetic interaction landscape between RNA binding proteins in maintaining organismal health, and will serve as a paradigm applicable to other biological questions.

***For correspondence:** john.calarco@utoronto.ca

**Competing interests:** The authors declare that no competing interests exist.

## Introduction

RNA binding proteins are critical regulators of all aspects of RNA metabolism, and have been implicated in human health and disease (*Castello et al., 2013*; *Gerstberger et al., 2014a*; *Lukong et al., 2008*). A number of studies have demonstrated the importance of specific combinations of RNA binding proteins for tissue-specific regulation of RNA processing and abundance (*Brooks et al., 2015*; *Chen and Manley, 2009*; *Elkon et al., 2013*; *Schoenberg and Maquat, 2012*). However, despite their importance in regulating many facets of gene expression, hundreds of metazoan RNA binding proteins have no described molecular or cellular function (*Castello et al., 2013*; *Gerstberger et al., 2014b*). Part of the challenge in characterizing these factors may be due to redundancy in their regulation of shared RNA targets. For practical reasons, genetic analyses in multicellular animals have been limited to ablating RNA binding protein genes in isolation (*Barberan-Soler et al., 2011*) or studying a single genetic interaction by targeting two members of the same family (*Gehman et al., 2012*; *Ule et al., 2006*). On the other hand, deletion of multiple RNA binding protein genes with overlapping functions should reveal stronger 'synthetic' phenotypes relative to the loss of individual factors, and shed light on the role of these genes in animal development and physiology. To systematically test this hypothesis, we sought to generate loss-of-function mutations in a defined set of RNA binding proteins in the nematode *Caenorhabditis elegans*, and then create a previously unprecedented number of double mutants, covering all possible pairwise combinations within that set.

In budding yeast and bacteria, the development of such systematic pairwise gene deletion schemes, including the synthetic genetic array (SGA) approach, have revealed widespread synthetic interactions between genes, in which double mutant phenotypes differ from what would be expected given observed single mutant phenotypes (*Butland et al., 2008*; *Costanzo et al., 2010*; *Schuldiner et al., 2005*). However, analogous genetic strategies in multicellular organisms have not been feasible. In *C. elegans*, RNA interference (RNAi) triggered by feeding animals double-stranded RNA has been used as a tool for analyzing genetic interactions, either by subjecting loss of function mutants to RNAi by feeding, or by targeting two genes simultaneously by RNAi (*Baugh et al., 2005*; *Byrne et al., 2007*; *Dixon et al., 2009*; *Lehner et al., 2006*). However, despite its throughput, the use of RNAi in genetic interaction studies can be limited since gene activity is knocked down rather than knocked out (*Kamath and Ahringer, 2003*), complicating the interpretation of synthetic phenotypes. Additionally, neurons are generally refractory to RNAi by feeding, unless specific hypersensitive genetic backgrounds are used which often have negative effects on animal physiology (*Asikainen et al., 2005*; *Calixto et al., 2010*; *Firnhaber and Hammarlund, 2013*; *Sieburth et al., 2005*).

Here, we have devised a strategy that overcomes these technical limitations by enabling the generation of null mutations in *C. elegans* that can be easily crossed together to create double mutants. We used this approach to study genetic interactions between neuronally-expressed RNA binding protein genes and found widespread interactions negatively impacting organismal fitness. One such genetic interaction between *exc-7* and *mbl-1*, two highly conserved RNA binding protein genes, led to a striking decrease in lifespan. Further analysis revealed that these genes potentially function downstream of the *daf-2* insulin receptor gene, a critical determinant of lifespan. Our results indicate that combinations of neuronal RNA binding proteins play a previously-unappreciated role in maintaining organismal fitness.

## Results

### Efficient CRISPR/Cas9-mediated double mutant generation in a multicellular animal

To systematically study genetic interactions using null alleles in *C. elegans*, we developed a strategy we will refer to as CRISPR/Cas9-based Synthetic Genetic Interaction (CRISPR-SGI) profiling (*Figure 1*). This strategy employs the CRISPR-associated enzyme Cas9 to generate double-strand DNA

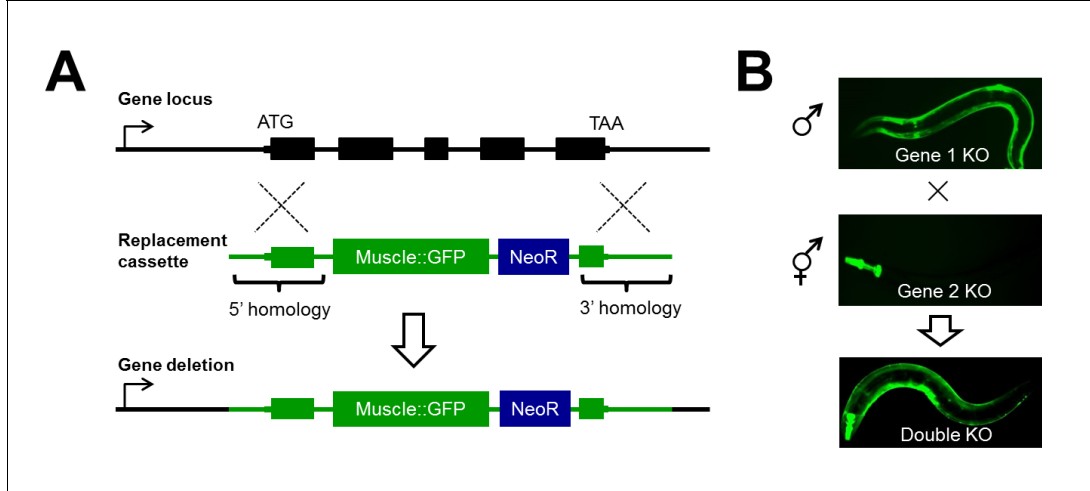

**Figure 1.** Disruption of RNA binding proteins via targeted CRISPR/Cas9-mediated homologous recombination. (**A**) Schematic of template-directed homologous recombination to disrupt a gene of interest and replace with a positive-selection cassette (NeoR = Neomycin resistance) and one of two tissue specific fluorescent marker genes labeling the pharynx or body wall musculature. (**B**) Double mutants are created by simple crosses on the fluorescent dissection microscope looking for presence of both fluorescent markers.

breaks in the genome (*Dickinson and Goldstein, 2016*; *Norris et al., 2015*). Following cleavage of chromosomal DNA by Cas9 and a single guide RNA (sgRNA), animals are created carrying marked deletion alleles through homology-directed gene replacement of a target gene with a heterologous GFP transgene. Thus, reporter fluorescence can be followed as a proxy for the gene deletion (*Figure 1A*). To make an array of double mutants, a collection of genes are independently replaced with two compatible non-overlapping tissue-specific GFP reporters, and then crossed to generate double heterozygous F$_1$ progeny (*Figure 1B*). These animals are allowed to self-fertilize and generate F$_2$ animals, and homozygous double mutants are isolated by microscopy. This strategy for creating double mutants provides a substantial increase in throughput as it eliminates the need for laborious large-scale PCR and/or sequencing to ascertain the genotype of the animal at each generation.

Using this technique, we focused on generating deletions in a set of 14 evolutionarily conserved RNA binding protein genes with expression in the nervous system, reasoning that RNA binding proteins co-expressed in the same tissue are more likely to exhibit synthetic phenotypes. These genes were selected by combining expression data from previously published studies with tissue-specific RNA sequencing we performed (*Supplementary file 1*). This list includes RNA binding proteins with well-known roles in the nervous system and other tissues, as well as factors with little or no known phenotype or function.

## Competitive fitness assays reveal fitness defects in RNA binding protein mutants

We created deletions marked by GFP repair transgenes in all 14 genes. Resulting phenotypes were consistent with previous literature, including two genes with strong neuronal phenotypes (*unc-75*/ CELF and *mec-8*/RBPMS2) (*Brenner, 1974*; *Chalfie and Sulston, 1981*; *Lundquist and Herman, 1994*) and one with a mild non-neuronal phenotype (*exc-7*/ELAVL)(*Fujita et al., 2003*) (*Figure 2— figure supplement 1*). To test for more subtle fitness defects, we took advantage of the fact that our mutants are labeled with GFP and designed a competition assay in which equal numbers of mutant and wild-type animals were placed on the same growth plate and allowed to grow for two generations. The fraction of GFP-positive worms on the plate at the end of the assay reflects the multi-generational competitive fitness of the mutant worms relative to wild-type worms (*Figure 2A* and see Materials and methods).

Performing the competition assay on single mutants revealed that while the majority of deletions had little or no effect on fitness, there were a few deletions causing substantial defects (*Figure 2B*). These included *unc-75* and *mec-8*, mutations in which are known to cause serious organismal phenotypes as described above (*Figure 2—figure supplement 1*) (*Brenner, 1974*; *Chalfie and Sulston, 1981*). Surprisingly, however, we also found that disruption of two genes, *C25A1.4* and the RBM24 ortholog *sup-12*, also led to significant defects in fitness. Interestingly, previous genetic studies had identified alleles of *sup-12* with no obvious visible phenotypes (*Anyanful et al., 2004*). Upon further analysis, we found that the *sup-12* fitness defects can be partially attributed to reduced growth rate, larval lethality and reduced fertility (*Figure 2C and D*). Importantly, these phenotypes were corroborated by a previously-unstudied *sup-12(ok1843)* deletion allele generated by the *C. elegans* knockout consortium (*Figure 2C and D*). We therefore believe that our deletion reveals the true null *sup-12* phenotype, while the previously-studied missense and splice-site alleles were partial loss-of-function alleles.

## Extensive synthetic fitness defects in RNA binding protein double mutants

Having created all single deletion mutants, we next generated all possible pairwise combinations of double mutants for thirteen of the fourteen mutants (*sup-12* was excluded due to difficulty in obtaining viable homozygous lines), for a total of 78 double mutants, representing the largest targeted collection of metazoan double mutants created in a single study to date (*Figure 3A*). Intriguingly, one of these mutants, *mec-8; exc-7*, could not be created as 100% of double homozygotes were embryonic lethal, although strains homozygous for one mutation and heterozygous for the other were viable (*Figure 3A–D*). This indicates that simultaneous loss of *mec-8* and *exc-7* RNA binding proteins leads to synthetic lethality. We further confirmed this synthetic lethality using the previously-

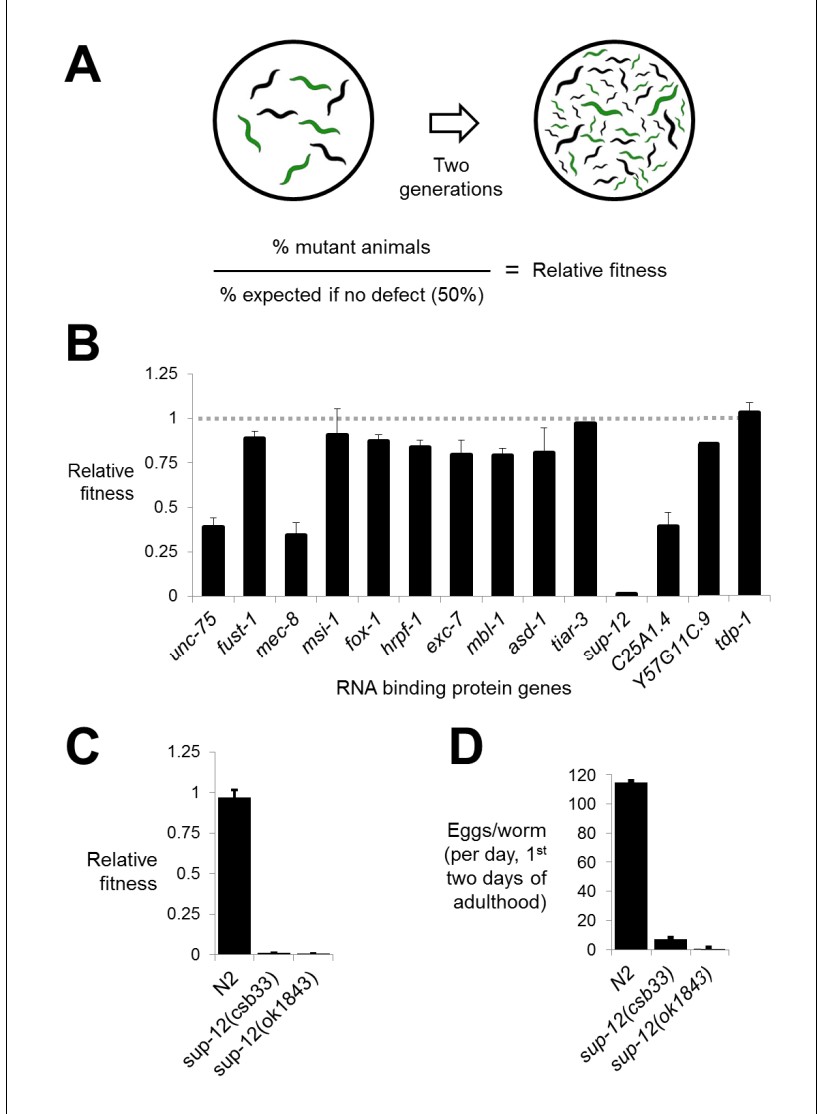

**Figure 2.** Competitive fitness assays identify RNA binding proteins with important roles in organismal fitness. (**A**) Competition assay schematic and calculation of relative fitness as fraction of mutant worms on plate divided by the null expectation of 0.5 assuming no fitness defects in the mutant. (**B**) Relative fitness values for all single RNA binding protein mutants. (**C**) Strong fitness deficit in *sup-12* mutants confirmed via independent *sup-12(ok1843)* allele. (**D**) Egg laying assay reveals fitness defects in *sup-12* mutants can be partially explained by greatly reduced fertility. Error bars = S.E.M.

The following figure supplement is available for figure 2:

**Figure supplement 1.** CRISPR mutants recapitulate known RNA binding protein phenotypes.

published canonical null alleles for *mec-8* and *exc-7* (*Chalfie and Sulston, 1981*; *Fujita et al., 2003*) (*Figure 3—figure supplement 1*).

We next searched for non-lethal synthetic effects on organismal fitness using the competitive fitness assay, performing several hundred competition experiments (*Supplementary file 2*). Starting with the relative fitness scores for all single mutants, we calculated predicted relative fitness values for the double mutants. The predicted values are based on null models (*Baryshnikova et al., 2010*; *Collins et al., 2006*; *Mani et al., 2008*) where for non-interacting gene pairs the expected relative fitness value of double mutants is the product of the two relative fitness values of each single mutant

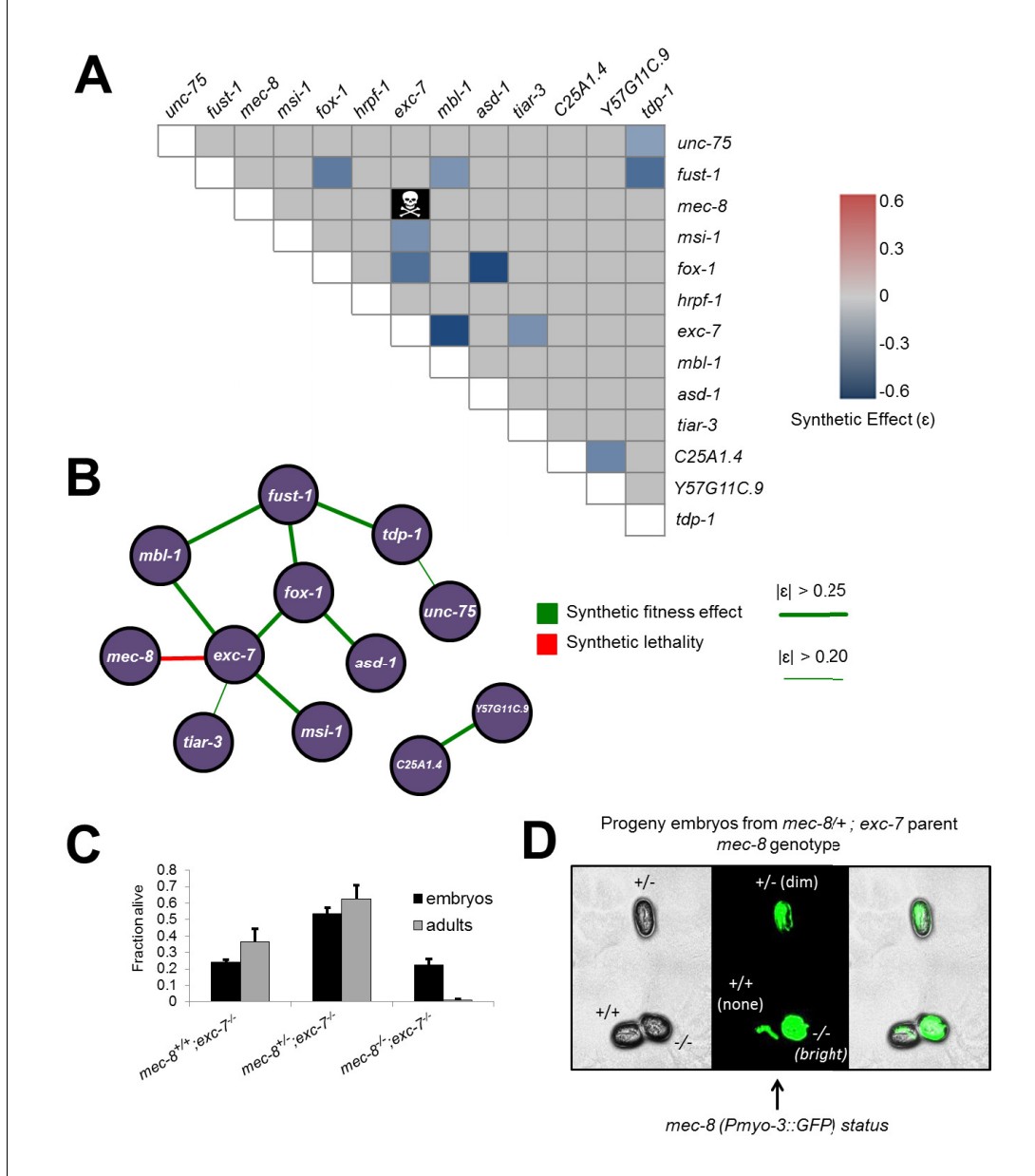

**Figure 3.** Extensive genetic interactions between RNA binding proteins affecting fitness, viability and lifespan. (A) Heatmap of synthetic effects for all pairwise genetic interactions (|ε| > 0.2). Skull and crossbones denotes synthetic lethality. (B) Network map demonstrating interconnectivity of genetic interactions among RNA binding proteins. Thickness of green edges denotes extent of measured fitness effect. (C) mec-8; exc-7 mutant embryos are laid from mec-8+/−;exc-7 mothers at expected Mendelian ratios, but double homozygotes die during embryogenesis. Error bars = S.E.M. (D) 'in situ' genotyping demonstrates that double homozygous embryos are laid at Mendelian ratios. Pharyngeal GFP marks homozygous exc-7 mutation, while body-wall muscle GFP brightness reports on the number of mec-8 mutant alleles (bright = −/−, dim = +/−, none = +/+).

The following figure supplements are available for figure 3:

**Figure supplement 1.** mec-8; exc-7 mutants are synthetic embryonic lethal.

**Figure supplement 2.** Calculation of synthetic effect (ε) and validation of lifespan defects.

(*Figure 3—figure supplement 2*). All double mutants were then competed against wild-type, and any strains showing a fitness difference (synthetic fitness effect $|\varepsilon| > 0.2$) from the predicted value were categorized as synthetically interacting (*Figure 3A*, *Supplementary file 2* and see Materials and methods).

The competition assay revealed a number of synthetic fitness effects relative to the null expectation, on a continuum from weak to strong fitness defects. Including the synthetic lethality result above, we identified 11 significant double mutant fitness effects (*Figure 3A and B*). Intriguingly, our detected synthetic genetic interactions involve 12 of the 13 tested RNA binding protein genes (*Figure 3B*), revealing many previously unknown connections between these factors. Some genes, such as *tiar-3*, have only one weak genetic interaction partner, while others are highly connected hubs, such as *exc-7*, which has one synthetic lethal interaction and four other synthetic fitness interactions, meaning that 45% of tested genes had a synthetic genetic effect in combination with *exc-7*.

## *exc-7; mbl-1* double mutants have severely reduced lifespans

While the competition assay has great sensitivity in detecting fitness disadvantages, it cannot pinpoint the nature of the phenotype leading to these defects. To explore this question we further investigated one of the strongest negative fitness interactions, found in *exc-7; mbl-1* double mutants (*Figure 2A*, $\varepsilon = -0.58$). Upon more detailed analysis of these animals, we noticed that a substantial proportion of adults died prematurely (*Video 1*). Specifically, while either *exc-7* or *mbl-1* single mutants are healthy three days into adulthood, the majority of *exc-7; mbl-1* double mutants have perished (*Figure 4A*). This defect can be recapitulated using independently generated deletion or premature stop mutations (*Figure 3—figure supplement 2*; *Brenner, 1974*; *Fujita et al., 2003*), and can be efficiently rescued by over-expressing either RNA binding protein in the double mutant (*Figure 4A*).

The synthetic nature of the *exc-7; mbl-1* phenotype prompted us to investigate whether these two factors exhibit overlapping expression patterns in particular cell and tissue types. We generated animals harboring fosmid reporters expressing fluorescently-tagged translational fusion proteins for EXC-7 and MBL-1 under their native regulatory elements. We found that EXC-7 is expressed in a number of tissues including the nervous system, while MBL-1 is expressed almost exclusively in the nervous system, in accord with previous reports (*Figure 4B*) (*Fujita et al., 2003*; *Spilker et al., 2012*). Moreover, within the nervous system the two factors exhibit strikingly specific patterns of co-expression. For instance, in the ventral nerve cord both factors are present in cholinergic motor neurons but neither are present in the GABAergic motor neurons (*Figure 4C*). This co-expression pattern suggests these factors may function together in specific neuron types to regulate partially overlapping networks of RNA targets.

## EXC-7 and MBL-1 combinatorially control the expression of hundreds of genes and isoforms

To better understand the molecular targets of *exc-7* and *mbl-1* we performed whole animal transcriptome sequencing on stage-matched wild-type, single mutant and double mutant strains. Intriguingly, we identified differences in both steady-state mRNA abundance and alternative isoform usage across the various mutants relative to wild type animals, with the most severe differences found in *exc-7; mbl-1* double mutants (*Figure 5*, *Supplementary files 3* and *4*). For instance, double mutants possess nearly five times as many differentially expressed genes as *exc-7* mutants relative to wild type animals (100 versus 23, respectively, *Figure 5A and B*). Additionally, we observed a strong bias in the direction of expression changes in double mutants, where in more than 90% of cases,

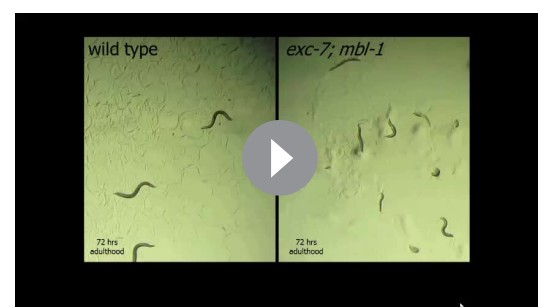

**Video 1.** Comparision of wild type and exc-7; mbl-1 double mutant animals.    A movie comparing the gross morphology and movement of wild type animals and *exc-7; mbl-1* double mutant animals at 72 hr post-adulthood.

expression levels are increased relative to wild type animals (*Figure 5A and C*). These results suggest that *exc-7* and *mbl-1* act together, either directly or indirectly, at the transcriptional or post-transcriptional level to inhibit excessive mRNA levels.

We also found that *exc-7* and *mbl-1* affected hundreds of alternative splicing events, many of which are annotated as being involved in lifespan regulation. Notably, a significant fraction (46%) of differentially spliced junctions controlled by *exc-7* were also controlled by *mbl-1* (*Figure 5D*, e, $p < 1 \times 10^{-60}$, hypergeometric test). Moreover, more than a hundred differentially spliced junctions were observed in *exc-7; mbl-1* double mutants that were not evident in either single mutant (*Figure 5E*). The most common classes of differential junction usage between wild-type and *exc-7; mbl-1* mutants were cassette-type exons and alternative 3' splice sites, which together accounted for over 50% of differential junction usage (*Figure 5—figure supplement 2A*). Focusing on cassette exons regulated by both RNA binding proteins, we searched for enrichment of biochemically-defined EXC-7 and MBNL1 *cis*-elements (*Norris et al., 2014*; *Ray et al., 2013*) in these exons and their flanking intron and exon sequences. Importantly, we observed enrichment of EXC-7 and MBNL1 *cis*-elements in the downstream intron of cassette exons undergoing increased skipping in the double mutants, and enrichment in the upstream intron of cassette exons undergoing increased inclusion the double mutants (*Figure 5—figure supplement 2B*). These results suggest that EXC-7 and MBL-1 regulate a network of alternative exons by binding directly to sequences in their surrounding introns.

Genes with expression changes in *exc-7; mbl-1* mutants exhibited no overrepresented Gene Ontology categories. On the other hand, genes with splicing changes in the *exc-7; mbl-1* mutants displayed numerous enriched Gene Ontology categories, indicating control over a diversity of important pathways (*Supplementary file 5*). Top enrichment categories include those associated with neuronal function, development and reproduction. This is consistent with the progressive defects observed in neuronal and reproductive function in *exc-7; mbl-1* double mutants (*Figure 6—figure supplement 1*). Collectively, these results indicate a high level of coordinate control of splicing networks between these two RNA binding proteins.

We confirmed a number of splicing changes between wild-type and mutant animals in lifespan-regulating genes via RT-PCR (*Figure 5F* and *Figure 5—figure supplement 1*), including two particularly interesting candidates. Specifically, we validated a significant alternative splicing difference in *unc-62*, a Meis/Homothorax family transcription factor whose alternative isoforms have been reported to differentially regulate lifespan (*Figure 5F*)(*Van Nostrand et al., 2013*). We also validated the differential usage of a 9-nucleotide micro-exon in the *unc-13/Munc13* gene, which was strongly regulated by both *mbl-1* and *exc-7* (*Figure 5F*). Intriguingly, *unc-13* is a well-characterized synaptic protein that has also been found to regulate lifespan (*Muñoz and Riddle, 2003*). These results suggest that *mbl-1* and *exc-7* coordinately modulate expression levels and a network of isoforms in genes including many responsible for governing longevity.

## *exc-7; mbl*-1 mutants inhibit *daf-2*-mediated lifespan extension

Collectively, our observations pointed to a phenotypic and molecular link between *mbl-1*, *exc-7* and lifespan. Thus, we analyzed the lifespan curves of these animals in greater detail. Interestingly, these assays revealed that premature lethality in *exc-7; mbl-1* animals is not apparent during development, but specifically in adulthood (*Figure 6A*). Moreover, developmental timing is normal, and behavioral correlates of organismal health such as locomotion and pharyngeal pumping are initially normal and only begin to decline in early adulthood (*Figure 6—figure supplement 1*). These results indicate that simultaneous loss of *exc-7* and *mbl-1* leads to dramatic effects on post-developmental health and longevity. Indeed, the *exc-7; mbl-1* lifespan defect is much stronger than in that of the classical shortened-lifespan mutant *daf-16/FOXO*, a conserved transcription factor and component of the insulin signaling pathway (*Figure 6B*) (*Lin et al., 1997*; *Ogg et al., 1997*). Additionally, exogenous expression of either *exc-7* or *mbl-1* transgenes significantly restored lifespan in double mutants (*Figure 6C*).

To determine whether these RNA binding proteins impinge upon known lifespan pathways or define a novel pathway, we performed epistasis experiments utilizing mutations in genes affecting insulin signaling (*daf-2*) or caloric restriction (*eat-2*). Both *daf-2* and *eat-2* mutants increase lifespan (*Kimura et al., 1997*; *Lakowski and Hekimi, 1998*) (*Figure 6D and E*) and we tested whether they were able to extend lifespan in the context of *exc-7* and *mbl-1* loss of function. Mutation of *eat-2*

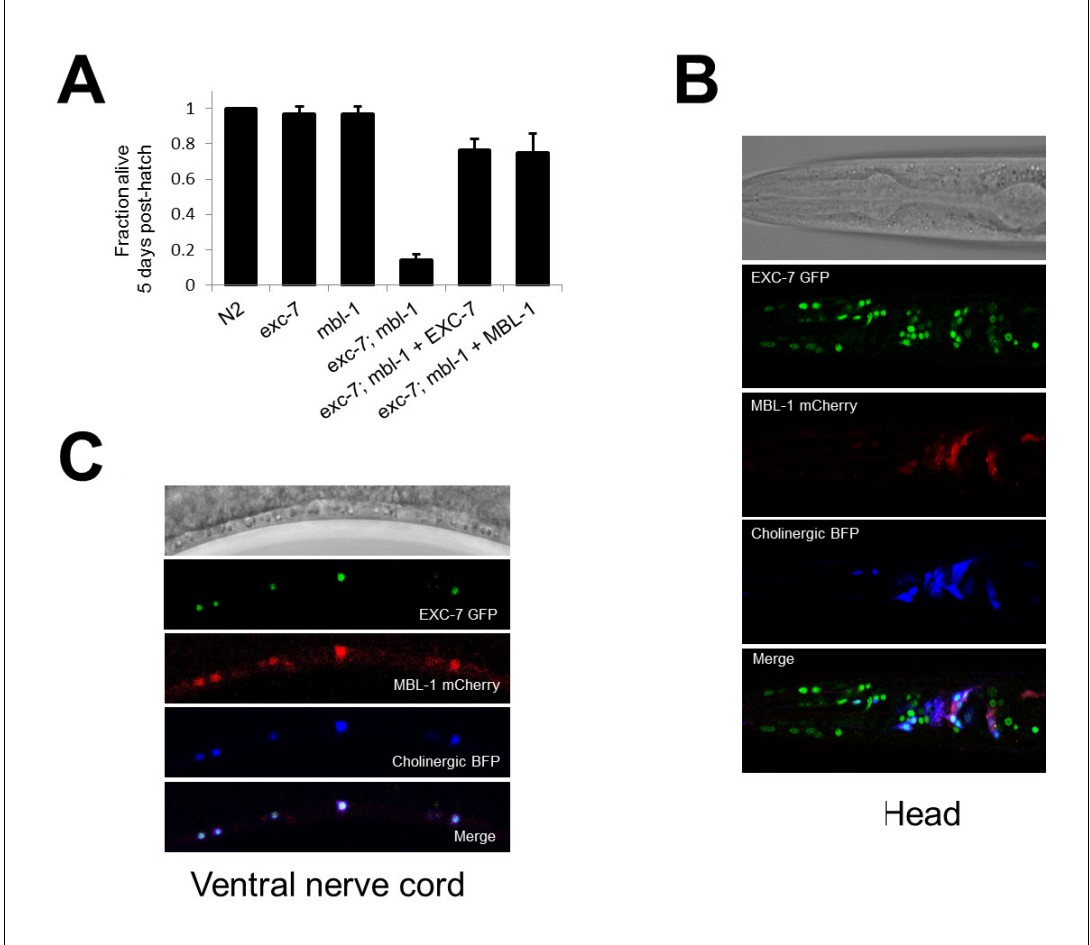

**Figure 4.** *exc-7* and *mbl-1* are co-expressed in specific neuronal subtypes. (A) exc-7; mbl-1 mutants die prematurely and can be rescued by overexpression of either EXC-7 or MBL-1. Error bars = S.E.M.(B) Head region of worm showing *exc-7* expression (GFP), *mbl-1* expression (RFP) and Cholinergic neurons (BFP). (C) Ventral nerve cord, demonstrating that both RNA binding proteins are co-expressed in the Cholinergic motorneurons.

increased lifespan in *exc-7; mbl-1* double mutants to a degree commensurate with its effect on wild-type worms (*Figure 6D*), indicating that *exc-7* and *mbl-1* are not necessary for lifespan extension by caloric restriction. Intriguingly however, mutation of *daf-2* failed to increase the lifespan of *exc-7; mbl-1* double mutants (*Figure 6E*), suggesting that the two RNA binding proteins may play a critical role in lifespan extension mediated by loss of insulin signaling.

## Discussion

### A robust method for systematic genetic interaction profiling in metazoans

In this study, we present a strategy for performing systematic synthetic genetic interaction screens in metazoans using null alleles for the first time. The strategy takes advantage of CRISPR/Cas9 technology, homologous recombination, and antibiotic selection to rapidly create transgenic animals. Heterologous fluorescent proteins are used to mark a gene deletion, thus allowing multiple deletion alleles with distinct heterologous fluorescent markers to be crossed together. Our approach has two immediate advantages over existing traditional methods of creating and screening double mutant animals. First, it enables the efficient generation of double mutants without the need for laborious PCR or sequencing validations. This offers a significant advance in throughput for generating double knockout mutants in genetically tractable multicellular animals. Second, because the approach

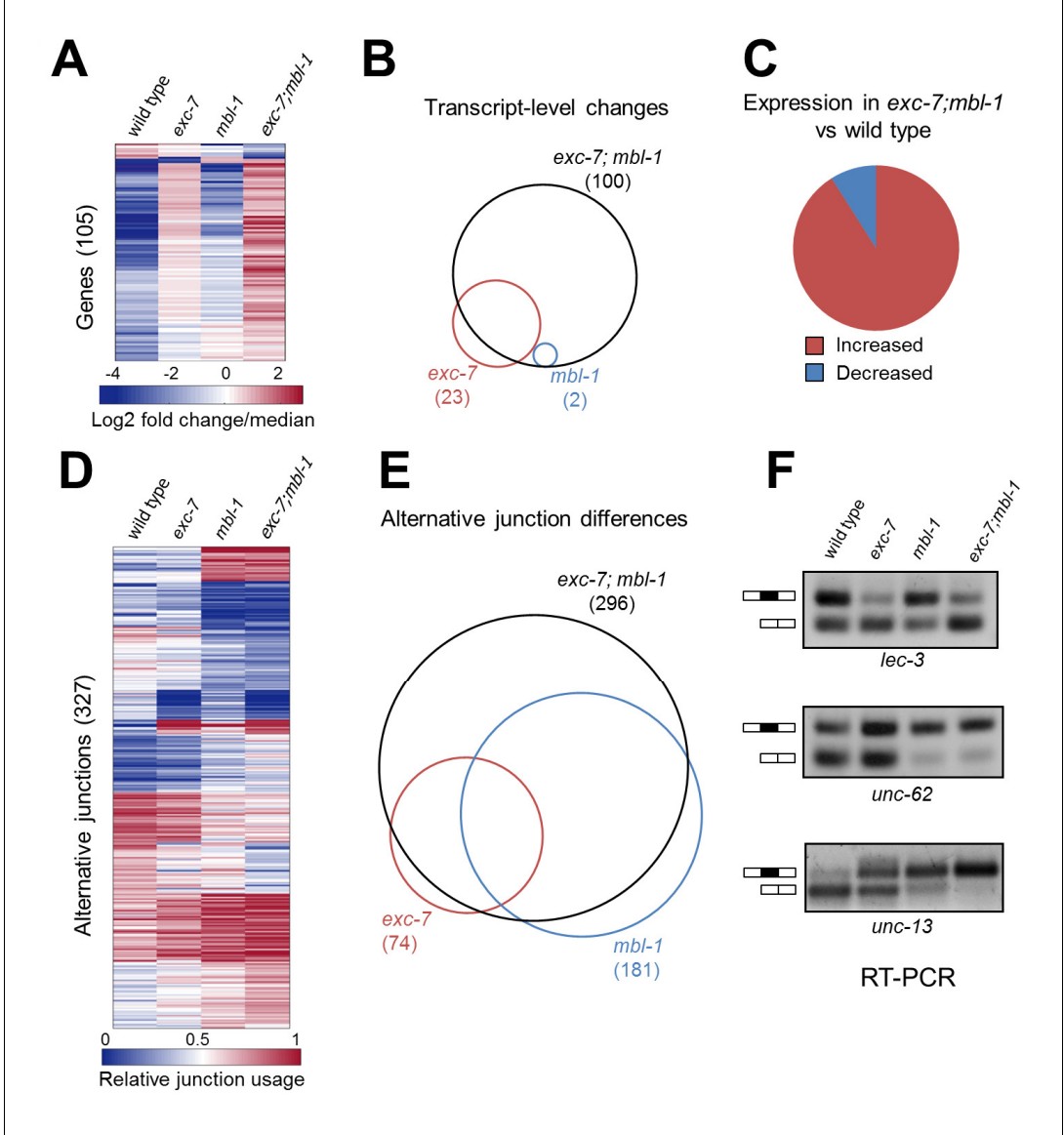

**Figure 5.** *exc-7* and *mbl-1* combinatorially control a large network of transcripts and alternative isoforms enriched for genes involved in lifespan regulation. (A) Heatmap depicting log2 fold change over mean values normalized across each row for wild type and mutant animal samples (columns). (B) Venn diagram depicting number of genes differentially regulated between wild type and mutant strains, and the extent of their overlap. (C) Pie chart depicting bias in expression level changes in *exc-7; mbl-1* double mutants relative to wild type animals. (D) Heatmap depicting relative alternative junction usage values for alternatively spliced junctions (rows) across wild type and mutant animal samples (columns). (E) Venn diagram depicting number of differentially regulated alternatively spliced junctions between wild type and mutant strains, and the extent of their overlap. (F) RT-PCR validations for candidate alternative exon skipping events showing changes between wild type and mutant animals. Upper bands represent exon-included isoforms and lower bands represent exon-skipped variants.

The following figure supplements are available for figure 5:

**Figure supplement 1.** Additional RT-PCRs validating splicing events detected in RNA-Seq analysis.

**Figure supplement 2.** Analysis of alternative splicing events regulated by EXC-7 and MBL-1.

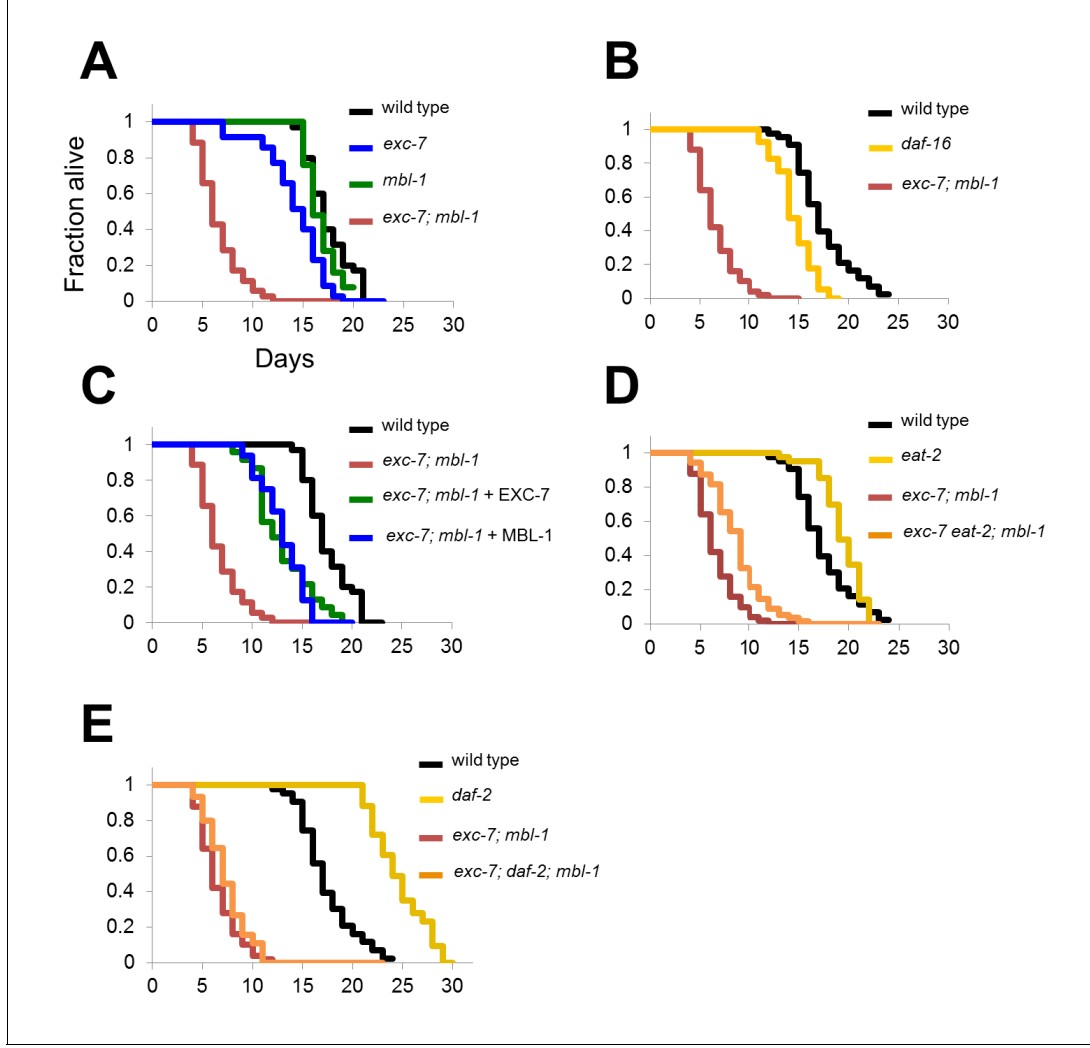

**Figure 6.** *exc-7; mbl-1* mutants have severely shortened lifespans, and genetically interact with the insulin signaling pathway. (A) *exc-7; mbl-1* double mutants, but not single *exc-7* or *mbl-1* mutants, have strongly reduced lifespans. (B) Lifespan deficits in *exc-7; mbl-1* are stronger than the classic short-lived mutant *daf-16(mu86)*. (C) Shortened lifespan in *exc-7; mbl-1* mutants can be rescued by transgeneic overexpression of either EXC-7 or MBL-1. (D) *eat-2(ad453)* mutation increases lifespan in wild-type worms and in *exc-7; mbl-1* mutants (p<0.001, log-rank test). (E) *daf-2(e1370)* mutation strongly increases lifespan in wild-type worms (p<0.001) but not in *exc-7; mbl-1* mutants (p>0.05).

The following figure supplement is available for figure 6:

**Figure supplement 1.** *exc-7; mbl-1* mutants appear normal throughout development but have severely shortened adult lifespans.

creates marked fluorescent mutant animals, these mutants can be competed with non-fluorescent wild type animals to screen for relative fitness under identical environmental conditions in a high throughput manner.

Although relative growth or fitness measurements appear superficial in nature, in budding yeast they have provided countless insights into functional connections between genetic pathways in the eukaryotic cell (*Butland et al., 2008*; *Costanzo et al., 2010*; *Schuldiner et al., 2005*). Analogously, we speculate that our competitive fitness assay will serve as a rapid screening tool to identify novel genetic interactions that can be characterized in greater detail (see below). However, customized screens could also be performed using variations of our gene targeting approach, taking advantage of the wealth of available molecular and organismal phenotypes known in *C. elegans*. This type of analysis is particularly amenable to *C. elegans*, due to its short generation time and hermaphroditic

reproduction. Such analysis in longer-lived, non-hermaphroditic species should be useful, but would require considerably more effort. We therefore believe our strategy will be broadly applicable to additional gene sets of interest, creating a blueprint for future synthetic interaction screens with null alleles in a multicellular animal.

## Widespread synthetic genetic defects among neuronal RNA binding proteins

We focused our CRISPR-SGI analysis on neuronal RNA binding proteins, a class of proteins previously shown to act in combinations and to be subject to synthetic genetic interactions (*Chen and Manley, 2009*; *Elkon et al., 2013*; *Schoenberg and Maquat, 2012*). Moreover, we reasoned that co-expressed gene families would be more likely to display genetic interactions. Our competitive fitness assays indicated that ~14% of all possible pairwise synthetic interactions resulted in a fitness change, and these interactions were distributed across >90% of all RNA binding protein genes tested. Interestingly, all detected genetic interactions in our study exacerbated defects in fitness, suggesting a general model where the combined loss of RNA binding protein genes leads to additive and/or synergistic effects on shared target transcripts (*Figure 7*).

This high interaction frequency represents a substantial enrichment in identified genetic interactions relative to a recent larger unbiased genetic interaction analysis reported in the budding yeast. Our observations thus reveal a high degree of genetic interaction among neuronal RNA binding proteins, and highlight the value of utilizing co-expression data in genetic interaction studies. Moreover, our competitive fitness assay is likely to miss subtler phenotypes such as mild behavioral defects, suggesting that the true degree of synthetic interactions among neuronal RNA binding proteins is likely to be even higher.

More generally, our results have important implications for neurological disorders involving RNA binding proteins (*Castello et al., 2013*; *Nussbacher et al., 2015*). Specifically, our data suggest that

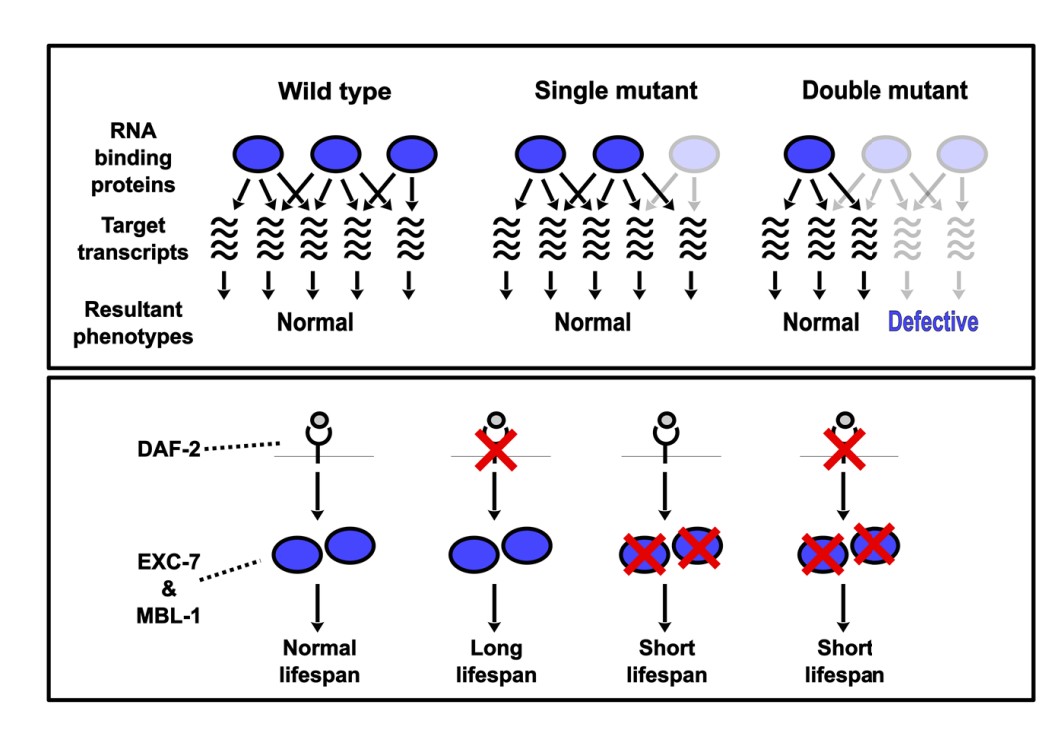

**Figure 7.** Model depicting synthetic genetic interactions of RNA binding protein genes in this study. Upper panel displays general model for observed aggravating synthetic genetic interactions between RNA binding protein genes, involving additive or synergistic effects on overlapping sets of target mRNAs. Bottom panel displays model based on our lifespan and epistasis experiments, placing EXC-7 and MBL-1 tentatively downstream of the DAF-2 insulin receptor in regulating lifespan.

mutations at additional RNA binding protein gene loci may frequently act as modifiers of the severity of these diseases. Indeed, it is well documented that mutations in several distinct RNA binding protein genes are associated with neurodegenerative diseases such as Amyotrophic Lateral Sclerosis (ALS) (*Nussbacher et al., 2015*; *Sreedharan et al., 2008*; *Vance et al., 2009*). Intriguingly, our double mutant screen has identified that *unc-75/CELF* displays synthetic interactions with both *tdp-1/TDP-43* and *fust-1/FUS*, two genes that have been implicated in ALS (*Sreedharan et al., 2008*; *Vance et al., 2009*). Additional work will be required to determine the nature of these genetic interactions we have observed in *C. elegans*, and whether the *CELF* family of RNA binding proteins act as modifiers of disease state in mammalian ALS models.

## Combinations of RNA binding proteins controlling lifespan

The promise of synthetic genetic interaction profiling to identify surprising phenotypes and unexpected combinatorial interactions is illustrated by the finding that *exc-7; mbl-1* double mutants have strongly reduced lifespans. While extensive work has been done to identify genes controlling metazoan lifespan, RNA binding proteins have been largely under-studied. However, recent evidence has emerged that the splicing factor SFA-1 is required for lifespan elongation mediated by dietary restriction in *C. elegans* (*Heintz et al., 2017*). Our findings extend this phenomenon to include an additional pair of RNA binding proteins, which we currently speculate are exerting a strong influence on lifespan by possibly acting downstream of the insulin signaling pathway (*Figure 7*). However, additional experiments are required to confirm the predictions of this model, including testing whether over-expression of these RNA binding proteins together have the ability to increase lifespan. Our results expand the spotlight on RNA binding proteins as important lifespan regulators, which have until now may have gone unrecognized due to genetic redundancy and/or combinatorial control.

It is intriguing to note that two RNA binding proteins co-expressed solely in the nervous system are together responsible for an aging phenotype that affects multiple tissues coordinately. It remains to be determined what the causal cellular or molecular mechanisms are linking the RNA binding proteins to organismal lifespan, but transcriptional profiling experiments indicate that the two factors combinatorially control hundreds of genes and isoforms, many of which are involved in mediating lifespan. We have found that EXC-7 and MBL-1 co-regulate gene expression in a 'synthetic' manner wherein expression is normal in either single mutant but aberrant in the double mutant. It is possible that such effects on steady state RNA levels could be the result of these factors regulating transcription or RNA stability, either through direct interactions with mRNA targets, or indirectly through the modulation of factors that control these processes. Indeed, it has been previously reported that both of these RNA binding proteins can influence other layers of RNA metabolism, including localization and RNA stability (*Simone and Keene, 2013*; *Wang et al., 2012*, *2015*). On the other hand, a large number of the splicing changes we observe are additive or cumulative in nature, wherein the double mutant splicing phenotype is reflective of each of the single mutant splicing phenotypes. It therefore may be that the double mutant synthetic effect on viability is caused by synthetic effects on RNA abundance. Alternatively, and not mutually exclusive, the synthetic phenotype may be caused by splicing events regulated additively and/or cumulatively by both factors, but only upon reaching a certain threshold of mis-splicing is a phenotype revealed. It will be of interest to identify the key targets of EXC-7 and MBL-1 that play crucial roles in maintaining long and healthy lifespan.

Collectively, we have developed a strategy for performing systematic genetic interaction analyses in metazoans using null alleles for the first time. Combining this approach with competitive fitness assays as a phenotypic readout led us to discover a hidden layer of novel genetic interactions. Our results thus highlight the importance of studying the effects of deleting combinations of RNA binding proteins in addition to studying single gene perturbations, and establish a novel paradigm in which these factors govern post-developmental health and longevity.

## Materials and methods

### Strain generation

*C. elegans* wild type N2 strain (RRID:WB-STRAINWB-STRAIN:N2_(ancestral)) was utilized for all CRISPR editing experiments. Mutants were created as previously reported (*Norris et al., 2015*).

Briefly, plasmids encoding Cas9, sgRNA, co-injection markers pCFJ90 and pCFJ104 (obtained from the Jorgensen lab through Addgene) and the selection cassette flanked by homology arms (500+ bp) were injected into wild-type worms. Insertions were identified by resistance to G418, loss of co-injection markers and uniform, dim fluorescence of the inserted GFP. Insertions were then validated by PCR amplicons flanking both upstream and downstream insertion sites, followed by Sanger sequencing to verify insertions. In most cases, mutants with complementary GFP markers were required, but in a few cases where single mutants had an obvious visual phenotype, it was possible for the sake of convenience to cross two mutations together marked by the same GFP marker. First the GFP marker for one gene deletion was homozygosed, and then the second gene with an obvious visual phenotype was homozygosed to create the double mutant. For co-expression analysis of *mbl-1* and *exc-7* genes (*Figure 3*), we generated an extrachromosomal array strain containing three co-injected constructs: an EXC-7::GFP fosmid, an MBL-1::SL2::mCherry fosmid (a gift from Kang Shen, Stanford University), and a Punc-17::BFP expression plasmid for labeling cholinergic neurons. For a complete list of strains generated in this study please see *Supplementary file 6*. Strain generation and fitness assays are described in greater detail at Bio-protocol (*Calarco and Norris, 2018*).

## Competitive fitness assay

Four L4 larvae of each genotype were picked onto standard NGM plates seeded with OP50 bacteria. Plates were incubated for five days at 25°C. For each biological replicate ≥150 total worms were counted. Fitness values and synthetic effect calculations were adapted from the yeast SGA literature which uses a multiplicative null model for discovering synthetic genetic effects (*Baryshnikova et al., 2010*). We define the Relative Fitness (F) for a mutant strain as the ratio of the observed percentage of animals on the plate divided by the percentage expected (50%) if the mutant had wild-type fitness levels: $F_1 = (100 \times (\# \text{mutant}_1 / \# \text{total worms}))/50\%$. For example, if 40% of worms on the competition plate are observed to be mutant, then the relative fitness value of the mutant relative to wild type is 40%/50%, or 0.8. Expected fitness for double mutants (Fexp) was the product of the two single mutant fitness values: $\text{Fexp}_{1,2} = F_1 \times F_2$. Synthetic fitness effects (ε) were the difference between observed (Fobs) and expected: $\varepsilon = \text{Fobs}_{1,2} - \text{Fexp}_{1,2}$. Conservative thresholds for significance were set at $|\varepsilon| \geq 0.20$. For strains passing the $|\varepsilon| \geq 0.20$ threshold, additional biological replicates were then performed. Finally, Fisher's exact test was applied between the aggregate observed values and the null-expectation values with a Bonferroni-corrected p-value of <0.01 used as significance threshold. Only genetic interactions passing all three significance criteria were reported as significant.

## Fluorescence microscopy

Animals carrying extrachromosomal arrays with broad expression were mounted on 2% agarose pads with 1x M9 and sodium azide, and imaged by fluorescence microscopy on a Zeiss LSM880 confocal microscope. Representative Z stacks were merged as maximum intensity projections and different channels were merged in Fiji.

## Lifespan assay

Staged L4 worms (n ≥ 100, split among three independent biological replicates) grown at 20°C were picked to NGM + FUDR (50 µg/mL) plates seeded with OP50 bacteria and grown at 20°C. Worms were considered dead when they ceased all spontaneous movement and no longer responded to touch from a platinum wire. Statistical significance was assessed by performing a log-rank test.

## Transcriptome sequencing (RNA-Seq) and computational analysis

Total RNA was extracted from L4 stage worms using Tri reagent (Sigma Aldrich) as recommended by the manufacturer. A total of three biological replicates were collected for each sample. PolyA+ transcripts were converted to cDNA libraries using the TruSeq RNA kit (Illumina). Sequencing generated 100 bp paired-end reads, which were mapped to the worm genome (versionWBcel235 using STAR)(*Dobin et al., 2013*). Gene-specific counts were tabulated for each sample using HT-Seq and statistically-significant differentially expressed transcripts were identified with DESeq (*Anders and Huber, 2010*). Differentially expressed transcript levels were then represented as log2 transformed fold change over mean values, and these values were clustered by Cluster 3.0 (http://bonsai.hgc.jp/~

mdehoon/software/cluster/software.htm) using K-means clustering (K = 10, 1000 runs). The resulting clustered data was used to generate a heatmap of the data presented in *Figure 4*.

For differential junction usage analysis, we obtained relevant splice junction counts from splice junction output files from STAR, and identified evidence of alternative junction usage by the following two criteria: (1) junctions that shared common start coordinates but had different end coordinates, and (2) junctions that shared common end coordinates but had different start coordinates. Relative junction usage values were calculated (junction1/(junction1+junction2) for all alternative junctions that had at least 30 counts across all samples, and then significantly different junction usage events between pairs of samples were identified by applying Fisher's exact test followed by a Bonferroni-adjusted p-value<0.05. The relative junction usage values for differentially spliced junctions were then used to cluster junctions into groups by K-means clustering (K = 10, 1000 runs) using Cluster 3.0. The resulting clustered data was used to generate a heatmap presented in *Figure 4*.

Proportional Venn diagrams were generated by inserting appropriate data points into the Bio-Venn GUI (http://www.cmbi.ru.nl/cdd/biovenn/)(*Hulsen et al., 2008*). Gene ontology analysis was carried out using AmiGo (*Carbon et al., 2009*) (GO Ontology database Released 2017-05-25) using Bonferroni correction. Motif analysis was performed on 50 cassette type exons (top 25 that were upregulated in the *exc-7; mbl-1* mutants, and top 25 that were downregulated). Since consensus cis-element data does not exist for the *C. elegans* MBL-1 protein, the consensus binding sites for the human homolog MBNL1 was used in its place. Top 10 7mer binding motifs for each RNA binding protein (*Ray et al., 2013*) were queried against cassette exons and their surrounding introns and exons, as well as an equal number of negative control exons and introns not affected by loss of either RNA binding protein. For our classification of frequency and types of alternative junction usage, we used proportions reported from (*Ramani et al., 2011*) as a wild type reference to compare differences in these distributions in our double mutant transcriptome data.

## Reverse transcription-PCR (RT-PCR) experiments

Candidate alternative splicing events were selected for independent validation by semi-quantitative RT-PCR validations. 20–50 ng of total RNA was used with gene-specific primers and the Qiagen OneStep RT-PCR kit as recommended by the manufacturer. Products were resolved on a 1.5% agarose gel stained with ethidium bromide. Images were acquired with an AlphaImager HP (Alpha Innotech).

## Acknowledgements

We would like to thank Ben Blencowe, Mei Zhen, Andrew Murray, Nicolas Chevrier, Arneet Saltzman, Joe Calarco, Jake Kirkland, and Megan Norris for valuable comments on this manuscript. ADN is supported by a Charles A King Trust postdoctoral fellowship. This research was supported by funding from NIH Early Independence Award grant DP5OD009153 and NSERC Discovery Grant 2017-06573 to JAC, from the Bauer Fellows Program (Harvard University), and from Canadian First Research Excellence Fund (Medicine by Design) and the University of Toronto. Some strains were provided by the CGC, which is funded by NIH Office of Research Infrastructure Programs (P40 OD010440).

## Additional information

### Funding

| Funder | Grant reference number | Author |
| --- | --- | --- |
| NIH Office of the Director | NIH Early Independence Award DP5OD009153 | John A Calarco |
| Harvard University | Bauer Fellows Program | John A Calarco |
| University of Toronto | | John A Calarco |
| Charles King postdoctoral fellowship | | Adam D Norris |
| Natural Sciences and Engi- | Discovery Grant RGPIN- | John A Calarco |

neering Research Council of        2017-06573
Canada

The funders had no role in study design, data collection and interpretation, or the decision to submit the work for publication.

## Author contributions

ADN, Conceptualization, Data curation, Formal analysis, Validation, Investigation, Visualization, Methodology, Writing—original draft, Writing—review and editing; XG, Resources, Data curation, Methodology, Writing—review and editing; JAC, Conceptualization, Formal analysis, Supervision, Funding acquisition, Investigation, Writing—original draft, Project administration, Writing—review and editing

## Author ORCIDs

Adam D Norris, http://orcid.org/0000-0002-0570-7414
John A Calarco, http://orcid.org/0000-0002-2197-7801

# Additional files

## Supplementary files

• Supplementary file 1. Genes deleted in this study. A table displaying the names of the *C. elegans* genes deleted in this study and their corresponding human orthologs.

• Supplementary file 2. Summary of competition experiments. Table displaying the results of all competition assays performed in this study. First worksheet contains single deletion mutant results, and second worksheet contains double mutant results.

• Supplementary file 3. Summary of differentially expressed genes in *exc-7* and *mbl-1* single mutants, and *exc-7*; *mbl-1* double mutants. Table displaying the results of our DESeq analysis identifying genes in mutants that are significantly differentially relative to wild type animals.

• Supplementary file 4. Summary of differentially utilized alternative junctions. Table displaying results of our alternative junction analysis comparing wild type, single mutant, and *exc-7*; *mbl-1* double mutant animals. Splice junction chromosome, start and stop coordinates, read counts, junction usage ratios, and p values are listed in the table.

• Supplementary file 5. Summary of Gene Ontology (GO) analysis. Table depicting the lists of statistically enriched GO terms associated with different comparisons to wild type animals (single and double mutant comparisons labeled in spreadsheet). The first worksheet contains analysis on differentially expressed genes, and the second worksheet contains analysis of differentially utilized junctions.

• Supplementary file 6. Strains used in this study. Table displaying information relevant to the mutant animals generated and used in this study. Allele identifiers, genotypes, regions spanning the deletions, and sgRNA targeting sequences used are all listed.

## Major datasets

The following dataset was generated:

| Author(s) | Year | Dataset title | Dataset URL | Database, license, and accessibility information |
|---|---|---|---|---|
| Adam D Norris, Xicotencatl Gracida, John A Calarco | 2017 | transcriptome analysis of wild type C. elegans and exc-7, mbl-1, and exc-7; mbl-1 mutants. | https://www.ncbi.nlm.nih.gov/bioproject/PRJNA386174 | Publicly available at NCBI (accession no: PRJNA386174) |

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
