## [Decision Letter]

Thank you for submitting your article "CRISPR-mediated genetic interaction profiling identifies RNA binding proteins controlling fitness and longevity" for consideration by *eLife*. Your article has been favorably evaluated by Aviv Regev (Senior Editor) and three reviewers, one of whom, Douglas Black, is a member of our Board of Reviewing Editors. The following individual involved in review of your submission has agreed to reveal their identity: Hidehito Kuroyanagi (Reviewer #2).

The reviewers have discussed the reviews with one another and the Reviewing Editor has drafted this decision to help you prepare a revised submission.

Summary:

This paper from Calarco and colleagues presents a method for genetic interaction screening in *C. elegans*. The authors create CRISPR-targeted knockout alleles. Each knockout is created as two alleles that yield expression of GFP in either pharynx or body wall muscle. Crossing lines carrying the pharynx-GFP mutation of one gene and the body-GFP of another gene, double mutants are rapidly identified by GFP expression in both locations. Single knockout mutants for 14 RNA binding proteins expressed in the nervous system were analyzed for fitness by competitive growth against wildtype worms. Only one mutation yielded a severe defect. The other 13 were all crossed with each other to characterize all possible double mutations. This uncovered 11 synthetic interactions showing a larger effect on fitness than the product of the two single mutants. One of these was lethal, with the others showing a range of severity. RNA binding proteins are an interesting test case for this method as they are known to act in particular combinations but generalized methods for testing for their cooperativity are lacking. The screen identified multiple interesting interactions for follow up, and the authors focused on one pair, *mbl-1* and *exc-7*, that exhibit a particularly large synthetic effect. These proteins have different patterns of cell specific expression but overlap in certain cholinergic neurons. The authors show that *mbl-1/exc-7* double mutant worms mature to adulthood in equal numbers to wildtype but then quickly exhibit reduced viability. RNA profiling of worms with the single and double mutant genotypes identified a large set of expression and splicing changes in the *mbl-1/exc-7* double mutants that are not seen in the singles. Given the observed shorter lifespan, the authors then examine the interaction of the *mbl-1/exc-7* mutant with various lifespan mutants. They find that the *eat-2* mutation affecting the caloric restriction pathway can extend the lifespan of the *mbl-1/exc-7* mutant in a similar manner to its effect on wildtype. In contrast, the *daf-2* mutation, which affects insulin signaling and which extends lifespan of otherwise wildtype worms, does not extend the lifespan of the *mbl-1/exc-7* mutant. From this they conclude that the effect of *mbl-1/exc-7* is downstream of *daf-2*.

All the reviewers agreed on the value of this study. A method to identify synthetic interactions between genes in metazoans is much needed. The authors make clever use of the worm system to develop a screen for such interactions. The method seems facile and adaptable to many other questions. The authors' examination of RNA binding protein combinations is also interesting. These proteins are understudied in the worm and this new method should allow questions to be examined that are hard to approach in other metazoan systems. The paper is well written and the data are clear. Several analyses need more detail, and a number of editorial changes are needed before it is ready for publication.

Essential revisions:

The authors relate the *mbl-1/exc-7* mutant to the lifespan mutations *daf-2* and *eat-2*, which increase lifespan. Mutations that increase lifespan would seem to be quite different from the many possible mutations that decrease lifespan. While formally epistatic, the loss of the *daf-2* effect in the *mbl-1/exc-7* mutant is not convincing evidence that the two are closely connected. Couldn't the *mbl-1/exc-7* worms just be dying before the *daf-2* has an effect? To really make this connection, shouldn't overexpression of *mbl-1* or *exc-7* increase lifespan? Or a mutation in one of the many targets that are upregulated in *mbl-1/exc-7*? This should be discussed and the claims and the title modified. Alternatively, they could show additional data supporting the claim that *exc-1* and *mbl-1* are involved in *daf-2* signaling. For example, does the *exc-1/mbl-1* double mutant suppress dauer formation phenotype of *daf-2*, in a similar manner to *daf-16*?

The authors demonstrated that overall expression of 100 genes and the splicing pattern in 296 genes are altered in the *exc-7; mbl-1* double mutant. More analysis should be presented of the kinds of genes affected. They should also discuss whether these transcripts are related in how they affect function of the nervous system, and especially cholinergic neurons and/or longevity. Specifically, if they examine the splicing or expression targets for gene ontology clustering (GO analysis), is there a clustering around insulin signaling or anything else? What is different between the single and double mutant target transcripts in their ontology enrichments? How might this relate to the synthetic lethality?

Similarly, they should report more detailed analyses of the exons affected by the *exc-7* and/or *mbl-1* mutations. Do they contain common regulatory elements for example? How do the splicing changes seen in the double mutant compare to the single mutants in terms of splicing pattern type (e.g. A5'SS, A3'SS, CE, MXE and IR)? Are there differences in these patterns that might explain the synthetic phenotype, or from the targets whose splicing pattern changes are seen only in the double mutant *mbl-1/exc-7* combination? (see Venn diagram – Figure 5).

Can the authors comment of why the overall gene expression profile is elevated by loss of RNA binding proteins and provide some plausible explanations?

In the supplementary data showing RT-PCR assays, it appears that, at least for the selected target exons shown, that there are few synthetic effects on splicing patterns. That is, one of the single mutants typically shows the same splicing pattern alteration as the double mutant. What does this mean for the double-mutant phenotype? Do they expect that the synthetic effect on viability would arise from a splicing target that also shows a synthetic effect or from combinations of targets that are changed by one of the two mutations?

---

## [Author Response]

*Essential revisions:*

*The authors relate the mbl-1/exc-7 mutant to the lifespan mutations daf-2 and eat-2, which increase lifespan. Mutations that increase lifespan would seem to be quite different from the many possible mutations that decrease lifespan. While formally epistatic, the loss of the daf-2 effect in the mbl-1/exc-7 mutant is not convincing evidence that the two are closely connected. Couldn't the mbl-1/exc-7 worms just be dying before the daf-2 has an effect? To really make this connection, shouldn't overexpression of mbl-1 or exc-7 increase lifespan? Or a mutation in one of the many targets that are upregulated in mbl-1/exc-7? This should be discussed and the claims and the title modified. Alternatively, they could show additional data supporting the claim that exc-1 and mbl-1 are involved in daf-2 signaling. For example, does the exc-1/mbl-1 double mutant suppress dauer formation phenotype of daf-2, in a similar manner to daf-16?*

We agree with the points raised by the reviewers regarding the epistasis experiments. Due to the complex and multi-faceted nature of lifespan extending versus shortening phenotypes, it may be difficult to draw strong conclusions from our epistasis analysis. We have included text in the manuscript that discusses some of the possibilities surrounding our observed phenotype. Additionally, we have altered the title of the manuscript to remove the word longevity and softened some of our statements throughout the text in relevant sections.

Anecdotally, we have not observed suppression of the dauer formation phenotype in *daf-2; exc7; mbl-1* triple mutants. These results would be consistent with the phenotype we observe that appears to manifest after animals reach adulthood, whereas the decision for animals to become dauers occurs earlier during larval development.

*The authors demonstrated that overall expression of 100 genes and the splicing pattern in 296 genes are altered in the exc-7; mbl-1 double mutant. More analysis should be presented of the kinds of genes affected. They should also discuss whether these transcripts are related in how they affect function of the nervous system, and especially cholinergic neurons and/or longevity. Specifically, if they examine the splicing or expression targets for gene ontology clustering (GO analysis), is there a clustering around insulin signaling or anything else? What is different between the single and double mutant target transcripts in their ontology enrichments? How might this relate to the synthetic lethality?*

We thank the reviewers for these suggestions. We have now performed GO analysis on both the genes with steady-state transcript level differences and genes with alternative junction usage differences. We did not find any molecular pathways or cellular processes that stood out as being enriched among the genes with steady-state level differences in double mutants. However, among the genes with differentially regulated alternative junctions, we identified a number of enriched GO terms associated with diverse aspects of development, reproduction, behaviour, and synaptic transmission and signaling. We have included this data in a new supplementary table, and we have also included text describing these results in the manuscript.

*Similarly, they should report more detailed analyses of the exons affected by the exc-7 and/or mbl-1 mutations. Do they contain common regulatory elements for example? How do the splicing changes seen in the double mutant compare to the single mutants in terms of splicing pattern type (e.g. A5'SS, A3'SS, CE, MXE and IR)? Are there differences in these patterns that might explain the synthetic phenotype, or from the targets whose splicing pattern changes are seen only in the double mutant mbl-1/exc-7 combination? (see Venn diagram – Figure 5).*

We have now classified the alternative junction usage events from our transcriptome analysis in double mutants, and present the relative frequencies of occurrence of each type of event and how these relative frequencies change relative to total documented frequencies of these events in *C. elegans*. We also performed an analysis of 50 cassette type alternative splicing events differentially regulated between wild type and double mutants (top 25 with increased inclusion, and top 25 with increased exon skipping in double mutants). Interestingly, we observe significant enrichment of both muscleblind-like and EXC-7 consensus motifs in the upstream and downstream introns flanking these alternative exons. We have included this analysis in a new figure supplement.

*Can the authors comment of why the overall gene expression profile is elevated by loss of RNA binding proteins and provide some plausible explanations?*

We agree that the bias towards increasing gene expression in the double mutants was a curious result. We speculate that these RNA binding proteins may be acting to regulate RNA stability in addition to their role in alternative splicing. Alternatively, these effects could be indirect via the regulation of one or more transcription factors or factors involved in controlling RNA stability.

We have included some additional discussion of these possibilities.

*In the supplementary data showing RT-PCR assays, it appears that, at least for the selected target exons shown, that there are few synthetic effects on splicing patterns. That is, one of the single mutants typically shows the same splicing pattern alteration as the double mutant. What does this mean for the double-mutant phenotype? Do they expect that the synthetic effect on viability would arise from a splicing target that also shows a synthetic effect or from combinations of targets that are changed by one of the two mutations?*

We agree with the reviewers’ assessment that a large number of splicing events altered in the double mutants are explained by changes observed in each of the single mutants. It is difficult to speculate whether the double mutant phenotype results from one or more aberrantly regulated targets that are synthetically controlled by the two RNA binding proteins, or the result of cumulative effects from independently de-regulated targets from each of the single mutants. We have included discussion suggesting that both of these scenarios are possible.